# Floristic Diversity and Distribution Pattern along an Altitudinal Gradient in the Central Andes: A Case Study of Cajatambo, Peru

**DOI:** 10.3390/plants13233328

**Published:** 2024-11-27

**Authors:** Yakov Quinteros-Gómez, Jehoshua Macedo-Bedoya, Victor Santos-Linares, Franco Angeles-Alvarez, Doris Gómez-Ticerán, José Campos-De la Cruz, Julio Solis Sarmiento, Abel Salinas-Inga, Zinnia Valencia-Saavedra

**Affiliations:** 1Laboratorio de Ecología Tropical y Análisis de Datos, Facultad de Ciencias Biológicas, Universidad Nacional Mayor de San Marcos (UNMSM), Lima 15081, Peru; jehoshua.macedo@unmsm.edu.pe (J.M.-B.); victor.santos2@unmsm.edu.pe (V.S.-L.); franco.angeles@unmsm.edu.pe (F.A.-A.); abel.salinas1@unmsm.edu.pe (A.S.-I.); zinnia.valencia@unmsm.edu.pe (Z.V.-S.); 2Facultad de Ciencias Matemáticas, Universidad Nacional Mayor de San Marcos (UNMSM), Lima 15081, Peru; dgomezt@unmsm.edu.pe; 3Herbario San Marcos, Universidad Nacional Mayor de San Marcos (UNMSM), Lima 15081, Peru; jocamde@gmail.com; 4Facultad de Ciencias Biológicas, Universidad Nacional Mayor de San Marcos (UNMSM), Lima 15081, Peru; jsoliss@unmsm.edu.pe

**Keywords:** Andean forest, Asteraceae, endangered flora, shrubland, vegetation

## Abstract

Introduction: This study focuses on the central mountain region of the Peruvian Andes, particularly its western slopes, high-altitude areas, and inter-Andean valleys situated above 3000 m.a.s.l. Despite its ecological importance, the region remains understudied, resulting in significant information gaps. Objective: To identify flora species along an altitudinal gradient in the Cajatambo district. Methods: Sampling was carried out at five distinct altitudinal levels using a combination of sampling techniques. Taxonomic identification was performed, and statistical analyses including ANOVA, the Mantel test, and NMDS were applied. Results: 424 plant species were identified, revealing the dominance of Asteraceae. The approach used allowed for the identification of floristic and structural patterns in various habitats, ranging from arid montane scrub to puna grassland. Surprisingly, Asteraceae richness had a significant impact on species diversity, while altitude did not. Additionally, floristic similarity between nearby altitudinal levels was not related to geographical distance. The analysis of ecosystems has shown that certain families are adaptable. Additionally, floristic diversity has been affected by human activity near the district capital. The distribution of medicinal species has been limited due to selective extraction. Conclusions: The shrubland and thorny scrub was the most diverse ecosystem and had the widest distribution across the altitudinal gradient.

## 1. Introduction

The Andes are one of the world’s richest regions in biodiversity, hosting a large number of endemic species of plants and animals. The formation of the Andes is attributed to complex tectonic processes that developed over millions of years [1], mainly the result of the convergence of the Nazca and South American tectonic plates [2]. The Andes mountain range covers almost all of western South America and cross seven countries: Venezuela, Colombia, Ecuador, Peru, Bolivia, Chile, and Argentina, and is characterized by its vast length of approximately 7000 km, making it the longest mountain range in the world [3]. This vast mountain system has a significant influence on the climatic patterns, hydrology, and biodiversity of the region [4], positioning itself as a fundamental factor in the configuration and maintenance of ecosystem services [5].

Peru, situated in the central–western region of South America, has an estimated 25,000 species of vascular plants [6]. The country also boasts an exceptional diversity of ecosystems, encompassing a wide variety of landscapes and natural environments [7] across diverse geographical regions, from the Pacific coast to the Amazon [8], and across the altitudinal gradient. The particular geography of the Peruvian landscape is accompanied by climatic variability, which has allowed the development of a diverse array of habitats [9]. The distinctiveness of the Peruvian flora is exemplified not only by the diversity of species, but also by the prevalence of endemic species [10]. These endemic species are frequently adapted to specific habitat conditions [11], making them crucial for the development of effective biodiversity conservation strategies; a total of 5509 taxa endemic to Peru are recognized, representing 27.9% of the country’s flora. [12].

The Central Mountain Region of Peru, as defined by Weberbauer (1945) [13], comprises the western slopes, the high Andean areas, and the inter-Andean valleys of the country (above 3000 m.a.s.l.). These areas have been the focus of interest for various researchers over time, among whom Hipólito Ruiz, José Antonio Pavón, August Weberbauer and Antonio Raimondi stand out [14,15], who have described them as sparse, discontinuous, and dispersed formations, highlighting their importance as unique reservoirs of biodiversity adapted to the specific conditions of these altitudes [16,17,18].

The Lima region, with a territorial extension of 35,892.49 km^2^, equivalent to approximately 3% of the Peruvian territory, is distinguished by its geographic and ecological uniqueness on the central Andean slope of Peru [19]. This area encompasses the provinces of Canta, Huarochirí, Yauyos, and Oyón, and reaches a higher elevation in Cajatambo, exhibiting a contrast of natural characteristics [20] along an altitudinal gradient. In this regard, Cajatambo exhibits herbaceous and shrub formations, with the sporadic presence of small trees [21]. These plants have evolved to thrive in cold and arid conditions [22], even in nutrient-poor soil, which limits the growth of woody species [23].

The central region of the Peruvian highlands has been the subject of historical exploration by renowned naturalists [24]. However, Cajatambo has been less extensively documented than other territories [25]. Even in the 20th century, the National Office for the Evaluation of Natural Resources (ONERN) conducted an inventory in the Oyón and Cajatambo basins in 1989, which reported a total of 74 plant species. In comparison, other provinces in the Lima region, such as Canta, have been the subject of more detailed floristic studies [26,27,28,29,30].

The objective of this study was to identify the flora species present in the Cajatambo district and to determine their distribution along an altitudinal gradient. Furthermore, the study aimed to evaluate the floristic similarity between altitudinal levels and their relationship with geographical distance.

## 2. Results

### 2.1. Floristic Inventory

A total of 424 plant species were recorded, grouped into 242 genera and 75 families (Table 1). Dicotyledons represent 79.6% of the taxa, with 337 species grouped into 56 families; monocotyledons represented 14.08%, with 60 species; Gnetophyta 0.23%, represented only by *Ephedra americana*; Equisetophyta 0.23%, represented only by *Equisetum bogotense*; and the remaining 5.86% represented by Pteridophytes, with 25 species included in 8 families.

The Asteraceae family was found to comprise 46 genera and 97 species (22.8%), with *Baccharis* and *Senecio* being the most diverse genera, with 11 and 16 species, respectively. Other frequently reported families included Poaceae (23 genera and 40 species), Fabaceae (15/27), Solanaceae (10/23), and Lamiaceae (11/20; Table 2).

Of the species identified, 69.5% exhibited an herbaceous habit, while shrubs were present in 18.3% of cases, trees and lianas in 3.4% each, and succulents in 4.5%. Epiphytes were only observed in 0.9% of instances (Figure 1). The families with the widest altitudinal distribution were also the most diverse, with Asteraceae being the most abundant at all altitudinal levels, especially above 3400 m.a.s.l. When analyzing Solanaceae and Fabaceae, it was found that the greatest number of species in these families was reported to be between 3400 and 3600 m.a.s.l., with their abundance decreasing as altitude increased. In contrast, the richness of the Poaceae family increased from 3800 m.a.s.l. This is due to the presence of several species belonging to the genera *Cinnagrostis* and *Poa* in this altitudinal range.

### 2.2. Ecosystems

In the months after the rainy season (April–May) a greening of all localities was observed, which is also related to the presence of a greater number of species, especially herbaceous species distributed in areas with water availability. The vegetation was characterized at each of the altitudinal levels (Figure 2).

(i) Level 1: Arid montane scrub in the town of Llocchi-Cañón (S = 23; elevation: 2600–2900 m.a.s.l.). This territory is characterized by being wild and dry (which justifies the limited anthropic activity), with temperatures that exceed 22 °C during the day. The Solanaceae family exhibited the greatest richness, representing 21.7% of species at the altitudinal level. The genus *Nicotiana* (*Nicotiana glutinosa*, *Nicotiana paniculata*, and *Nicotiana rustica*) demonstrated the highest diversity. The majority of species (70%) were herbaceous, followed by trees (17.4%) and shrubs (8.7%). The presence of some Cactaceae was also observed. The most prevalent tree and shrub species in this ecosystem were *Schinus molle* and *Mutisia acuminata*, respectively.

(ii) Level 2: The Cactaceae floor in the vicinity of the ravines and surrounding Cruzjirca (S = 202; elevation: 2900–3200 m.a.s.l.), where the Asteraceae family exhibited the greatest richness, representing 16.8% of plant species. The dominant species was *Flourensia macrophylla*, while the most diverse genera were *Conyza*, *Cronquistianthus*, and *Ophryosporus.* In this ecosystem, 47.4% of the species present in this study were reported. The majority of these were herbaceous (64.9%), followed by shrubs and succulents, which constituted 21.8% and 5.4% of the total, respectively. Cacti were observed in significant quantities in inaccessible areas, with a notable presence of *Browningia pilleifera*, *Cleistocactus fieldianus*, *Echinopsis pachanoi*, and *Opuntia ficus-indica*. Additionally, 18 species of legumes were identified, including *Spartium junceum*, *Caesalpinia spinosa*, *Otholobium pubescens*, and *Senna birostris*. In areas where water is available, we found crops of *Vicia faba* and *Medicago sativa*. In these territories, *E. pachanoi* is extracted for use in rituals.

(iii) Level 3: Shrubland and thorny scrub in the surroundings of Cajatambo (S = 337; elevation: 3200–3600 m.a.s.l.) in territories crossed by the Isco River, with little water in the dry season. The Asteraceae family was the most important, representing 21.4% of plant species, with the genera *Senecio* and *Baccharis* being the most diverse. Fabaceae was the second most diverse family, representing 8% of plant species, with the genus *Lupinus* being the most diverse. In this ecosystem, 79.1% of the species present in the district were found; again, herbaceous plants were dominant, at 66.6%. At this altitudinal level are most of the territories for agriculture and livestock, which also implies a greater anthropogenic impact on these territories. In areas with water availability, we found ferns, *Minthostachys mollis*, *Urtica magellanica*, *Equisetum bogotense*, and some Brassicaceae. In xeric areas, we found *Austrocylindropuntia subulata* to be dominant (Figure 3F), alternating with individuals of *Hesperomeles cuneata*, *Berberis flexuosa*, *Berberis lutea*, and *Duranta armata*, characteristic for the presence of spines.

(iv) Level 4: This is the transition territory from humid scrubland to grassland (S = 233; elevation: 3500–4000 m.a.s.l.) in the town of Rancas, situated at the foot of the snow-capped Huacshash (Huayhuash mountain range) in the upper basin of the Pumarrinri River. The area between 3500 and 3700 m.a.s.l. is characterized by the presence of small tributaries originating from the upper slopes of the mountain, which contribute to the availability of permanent water sources. The dominant vegetation comprises *Achyrocline alata*, *Baccharis chilca*, *Baccharis latifolia*, *Ambrosia peruviana*, *Minthostachys mollis*, *Urtica magellanica, Bomarea dulcis*, and individuals of the Calceolariaceae family. As we ascended, the vegetation became increasingly smaller, and we began to find individuals belonging to the Loasaceae family, as well as Asteraceae species such as *Paranephelius ovatus*, *Perezia multiflora*, and *Culcitium canescens*. It has been documented that between 3700 and 3800 m.a.s.l. there is a presence of *Polylepis* and *Puya alpestris* (Figure 4A). At elevations above 3800 m, the Poaceae become the dominant grass family, with the genera Cinnagrostis, *Festuca*, and *Calamagrostis* being the most diverse. In this territory, herbaceous plants were the most prevalent, representing 70.7% of individuals. Due to the challenging accessibility and distance to the district capital, this territory is the least affected by human activity, with some potato crops observed between 3500 and 3550 m. Additionally, the abandonment of homes was evident, indicating a significant depopulation of the territory.

(v) Level 5: Puna grassland in the upper part of the Cerro San Cristóbal hill and on the road to the Raura mine (S = 44; elevation: 4000–4400 m.a.s.l.). In this territory, the soil is covered with different species of grass (28 species). Indeed, the Poaceae family demonstrated the greatest richness, representing 59.1% of plant species, with the genera *Poa* and *Cinnagrostis* being the most diverse. Asteraceae was the second most diverse family, representing 27.3% of plant species; *Baccharis* was the most diverse genus (6 species). A total of 82.9% of species registered for this ecosystem correspond to herbaceous plants. Anthropic activity in this territory is almost non-existent, functioning as a transit territory towards Oyón and Huayllapa (entrance to the Huayhuash mountain range), regularly frequented by muleteers and tourists.

The generated graphs, which display both dendrograms and heat maps, yield interesting findings. The dendrograms illustrate the existence of similarities in the composition of botanical families on the Y-axis for the altitudinal gradient and the transects employed in the analysis, while the heat maps elucidate the abundance of species per family in the sampling area, thereby highlighting the disparities in their distribution along the altitudinal gradient (Figure 5).

### 2.3. Sampling Effort in Transects

The species richness for the transects, according to the non-parametric estimators (ICE, Chao2, Jackknife1, Jackknife2, and Bootstrap) was between 27 and 37 species. The non-parametric Bootstrap indicator was the one that registered the best fits based on the observed richness, with 89% of species reported. The ICE and Jackknife1 indicators also registered a good fit, with 78.5% and 75.6%, respectively, while Chao2 (72.7%) and Jackknife2 (64.1%) reported curves with less fit (Figure 6).

### 2.4. Inferential Statistics

The ANOVA revealed a significant effect of Asteraceae richness on transect richness along the altitudinal gradient (F = 7.67, df = 7, *p* = 0.0002), whereas altitude did not show a statistically significant effect (F = 2.11, df = 4, *p* = 0.1217). Additionally, the multiple regression analysis indicated a significant relationship among the variables analyzed at a 95% confidence level (F = 18.72, df = 2, *p* = 0.000). Furthermore, the regression model suggested that altitude alone does not significantly affect transect richness (T = −1.19, *p* = 0.2431), contrasting with the highly significant effect of Asteraceae richness (T = 5.36, *p* = 0.0000). Moreover, the Mantel Test found no significant correlation between geographical distances and floristic similarity (Jaccard index: R = −0.41, *p* = 0.236).

The NMDS illustrates the interrelationship between botanical families at each altitudinal level, with a stress level of 0.197 (Figure 7; ANOSIM: R = 0.6019, *p* = 0.0001). In the NMDS, an overlap of ellipses was observed between certain altitude levels. This was made possible by the considerable number and variety of families present at each level. Moreover, a close relationship was identified between levels 1 and 2 (blue and orange ellipses, respectively), with the shared presence of species from the families Anacardiaceae, Bignoniaceae, and Solanaceae. Additionally, an overlap was observed between the ellipses of levels 3 and 4 (red and green, respectively). These levels are primarily composed of families such as Cactaceae, Asteraceae, Poaceae, and Fabaceae. Finally, level 5 (purple ellipse) was observed to be slightly isolated, as it restricts the distribution of certain species belonging to the Asteraceae and Poaceae families.

### 2.5. Threatened Flora

In accordance with Decreto Supremo N° 043–2006–AG, 18 species of flora were identified as potentially threatened within the Cajatambo district (CR 17%, NT 44%, VU 39%). Of these, three are currently classified as critically endangered: *Polylepis racemosa*, *Buddleja coriacea*, and *Buddleja incana*. As indicated by the Red List of the International Union for Conservation of Nature (IUCN), 57 species of flora were identified as being subject to some degree of threat (DD 3.5%, EN 1.8%, LC 87.7%, NT 3.5%, VU 3.5%). Of these, *Juglans neotropica* was the sole endangered species, while *Polylepis racemosa* and *Polylepis weberbaueri* were classified as vulnerable (Figure 8).

## 3. Discussion

The Andean and sub-Andean territories have not been the focus of significant research regarding biodiversity, particularly in the Lima region [28,29]. Despite this, valuable information on the composition of the vegetation has been provided, with reports of 184 and 181 species, respectively [28,29,31]. However, knowledge of plant diversity in the Cajatambo district remains limited [32], with significant gaps in the available information, in particular in areas where there are no logistical facilities for development due to the remoteness of these wild territories and the topographic constraints posed by steep slopes. In this context, this research aims to address this knowledge gap by providing comprehensive information from sampling carried out in remote territories, encompassing a broader altitudinal gradient.

The altitudinal gradient evaluated revealed that Asteraceae, Poaceae, and Fabaceae were dominant, a result that is in line with previous studies in the same district [21,31]. The genera *Senecio* and *Baccharis* exhibited the greatest diversity. This pattern is similar to that reported in other provinces of Lima [19,28,33], as well as in other high Andean areas [34,35].

In terms of the plants’ growth habit, herbaceous plants were the most representative (approximately 70%). This is consistent with the previously mentioned research, as the study areas are grasslands with sparse vegetation with widely dispersed trees and shrubs [35]. The findings of the study by Cano et al. (2010) [36] in cryoturbated soils of Ancash at an elevation of over 4500 m.a.s.l. provide further evidence of the dominance of herbaceous plants, representing over 96% of the reported flora. The differences in plant growth habit observed along an altitudinal gradient are related to the composition of the vegetation and its forms [13,37]. In the puna grassland, herbaceous plants represent the majority of life forms, with the exception of some shrubs belonging to the *Baccharis* and *Buddleja* genera. These findings were corroborated by the observations of Weberbauer (1945) [13] and González (2016) [32], who note that the greatest diversity of Asteraceae is present between 2500 and 3500 m.a.s.l. By contrast, [28] found that the Asteraceae also exhibited dominance above an elevation of 4000 m, alternating with the Poaceae of the genera *Agrostis*, *Calamagrostis*, and *Poa*.

The Asteraceae family is distributed globally, with a cosmopolitan distribution. In Peru, it is the second most diverse plant family, after orchids [38,39]. This phenomenon can be attributed to the diverse pollination and seed dispersal mechanisms observed in Asteraceae [40], in addition to their adaptive characteristics, such as pubescence, flexible roots, and low physiological performance to survive low temperatures [41], which allow them to be favored by low turnover values between communities.

The Poaceae family has a distinctive and specialized photosynthesis process that, when combined with the action of wind pollination (anemophily), the structure of its inflorescences, and the formation of an indehiscent dry fruit (caryopsis), has enabled this family to adapt to a diverse range of terrestrial ecosystems. This includes cold territories [42,43] such as Cajatambo. Other families that exhibited a notable presence were Lamiaceae, Solanaceae, and Fabaceae. The first has a particular preference for temperate climates in mountainous areas [44], while Solanaceae presents a great multiplicity of vegetative and reproductive forms, which allows its adaptation to varied environments between sea level and the 4000 m elevation [45]. It is also noteworthy that the Fabaceae family comprises a vast array of species that have evolved to thrive in a multitude of habitats, contributing to its status as the third largest family in the world [46].

Non-parametric methods were employed to estimate species richness, as they are less biased compared to the species accumulation curve [47]. The Bootstrap and ICE estimators were the most effective at estimating species richness, producing highly representative results. In this context, the Bootstrap estimator is regarded as a highly reliable and robust method for estimating species richness when considering presence–absence data [48,49]. The ICE curve exhibited a pronounced initial growth phase, indicative of a random distribution rather than an aggregate one. The Chao 1 and ACE estimators were not employed in this study, as they are highly susceptible to aggregation [50]. The results of the ANOVA and multiple regression model indicate that the richness of Asteraceae species in the study area is regulated, contrary to the influence of altitude. It is commonly assumed that altitude is a limiting factor [40], yet this was not substantiated by the present investigation. This suggests that the adaptive characteristics of Asteraceae enable them to expand their distribution [51], leaving few opportunities for the establishment of other botanical groups. The Mantel test indicated that there was no relationship between geographical distance and similarity between transects. It was observed that the closest altitudinal levels were not necessarily the most similar floristically, indicating that proximity between transects does not necessarily influence the proportion of shared species.

The NMDS showed that the shrubland and thorny scrub share a large number of species with the transition territory from humid scrubland to grassland, which is why they are shown as a single assemblage. In the case of the arid montane scrub and the cactus floor around Cruzjirca, these only coincided in 50% of the transects, while the puna grass appeared as an independent ecosystem. These observations suggest that at an altitude of between 3300 and 4000 m elevation, the floristic groups are more homogeneous, sharing a greater number of species. Another factor to explain the floristic composition between transects is the ease of dispersal of Asteraceae and Poaceae, which can adapt to extreme conditions and directly influence their richness and abundance. On the contrary, the distribution and abundance of rare species can be affected by environmental barriers [52,53], as well as by anthropogenic activity in areas near the district capital, where livestock and agricultural exploitation has eroded floristic diversity. Indeed, the selective harvesting of medicinal species has left deforested areas where post-agricultural succession is still limited when compared to the richness at the altitudinal level.

It is evident that certain species, including *Echinopsis pachanoi*, *Baccharis latifolia*, *Minthistachys mollis*, *Perezia multiflora*, *Urtica magellanica*, and *Culcitium canescens*, have seen their distribution limited as a consequence of their selective extraction for the production of local preparations to combat different types of conditions [54,55]. It is of the highest importance to continue the study of floristic diversity in Cajatambo, extending efforts to the other districts and reaching an altitude of greater than 5000 m.a.s.l. This is with the purpose of reporting and describing little-known ecosystems where species with restricted distribution (endemic) and new to science could be found. In order to achieve this, it is essential to propose management and conservation tools that are linked to research projects. This will enable decision-makers to understand the complex ecological systems and to execute realistic and inclusive development plans where communities are involved [56]

## 4. Materials and Methods

### 4.1. Study Area

The name of the province of Cajatambo comes from the combination of two Quechua words: “Casha”, which means thorn, and “tampu”, which refers to “lodging” or “inn”. The combination of these two words forms the word “Cashatampu”. The district of Cajatambo, the provincial capital, is located at 3376 m.a.s.l. (10°28′21.4″ S and 76°59′34.6″ W; Figure 9) northeast of the Lima region in the western Andean Mountain range, with a maximum elevation of 6127 m on the snow-capped Sarapo mountain, in the upper basin of the Pativilca river.

In the lower parts of the district (around 1960 m.a.s.l.), the climate is temperate, with sporadic rainfall. As one ascends the slope, the temperature gradually decreases and rainfall becomes more abundant (rainy season from December to April [57]), reaching up to 1500 mm in the highest peaks [58].

Cajatambo is characterized by a population whose main occupation is agriculture and livestock. There is considerable poverty in the region, due to the inequalities created by exclusive land ownership and lack of specialization. This has led to a negative population growth rate of more than 20% between 2007 and 2017, making Cajatambo the least populated province in Lima.

### 4.2. Sampling Method

Fieldwork was carried out between July 2022 and June 2023 with the aim of recording the greatest number of flowering species. Five sampling plots were established in an altitude gradient from 2600 to 4400 m elevation. The road that connects the district of Cajatambo with Barranca and the town of Huayllapa was our main reference for carrying out the transects. In each sampling area (elevation level), six transects of 50 × 5 m were made, with an area of 0.15 ha, distributed in a directed manner (access and slope), for a total of 0.75 ha in 30 transects, using the method proposed by Gentry (1982) [23], with modifications by Mendoza (1999) [59] and Álvarez et al. (2001) [60]. Transects included trees and shrubs with a diameter at breast height (DBH) ≥ 1 cm and a height greater than 1 m, as well as herbaceous plants and lianas. Sampling was complemented by permanent surveys and walks in each of the sampling areas throughout the study period.

We collected botanical samples for taxonomic identification of species by comparison with herbarium specimens and specialist literature. Botanical nomenclature was based on W3-Tropicos (www.tropicos.org). The conservation status of recorded species was noted according to the Red List criteria and the Peruvian categorization of threatened flora species (Decreto Supremo N° 043–2006–AG).

### 4.3. Data Analysis

A community matrix was constructed with the original data, and heat maps were generated using the ggplot2 package of the RStudio software (version 2(1)) [61], with the aim of visualizing the distribution of families along the five altitudinal gradients. In addition, specific analyses were carried out within each gradient, using the transects studied in each of them.

Sampling efficiency was assessed for the 30 transects using species accumulation curves for the non-parametric estimators Jacknife1 and 2, Bootstrap, ICE, and Chao2 [62]. Non-parametric estimators are statistical tools designed to address the challenge of estimating species richness in a population, taking into account factors such as species presence/absence and frequency of occurrence [63]. According to Melo (2004) [64], these methods are based on the frequency of rare species in a sample to estimate total richness. Jackknife 1 considers species occurring in a single sample, while Jackknife2 also includes those occurring in two samples, and Chao2 is based on the occurrence of species in samples. Melo demonstrates that these estimators tend to closely follow the observed species accumulation curve, with Jackknife2 producing the highest estimates and Jackknife1 the lowest among these methods. Interestingly, the Jackknife estimators showed stronger correlations with the number of observed species than the Chao2. All estimators were calculated using EstimateS version 9.1 [65,66].

A two-way ANOVA was performed to determine the effect of richness of the most diverse family (Asteraceae) and altitudinal level on transect richness. Multiple correlation tests were also performed between Asteraceae richness, altitudinal levels, and transect richness.

Mantel tests based on Monte Carlo permutation methods were performed to assess whether there was a significant relationship between geographic distance and floristic similarity between elevation levels [67]. We use a data matrix with the Jaccard index (sensitive to sample size in assemblages with many rare species) and a distance matrix. The matrix correlation was estimated using a Monte Carlo analysis with 9999 permutations in XLSTAT version 2023.2.0.

Non-metric multidimensional analysis (NMDS) was used to evaluate the vegetation structure at the family level between altitudinal levels. An analysis of similarity (ANOSIM [68]) was performed to determine the significance of the NMDS. The NMDS and ANOSIM were performed with the R Studio program version 2023.09.1+463 using the vegan [69] and ggplot2 [70] packages.

## 5. Conclusions

This study successfully identified and characterized the floristic diversity of the Cajatambo district, documenting a total of 424 species distributed along an altitudinal gradient of 2600 to 4400 m.a.s.l. The results reveal complex distribution patterns, with the Asteraceae family dominating across all altitudinal levels, while other families, such as Solanaceae, Fabaceae, and Poaceae, exhibited variations in abundance and diversity according to altitude. Five altitudinal levels with distinctive vegetation characteristics were identified, ranging from arid montane scrub to puna grassland.

Contrary to expectations, no significant correlation was found between geographical distance and floristic similarity, as indicated by the Mantel test. NMDS analysis demonstrated some overlap in floristic composition between certain altitudinal levels, particularly between levels 3 and 4.

It is noteworthy that Asteraceae species richness did have a significant effect on overall transect richness, while altitude did not exhibit a statistically significant effect. These findings provide a deeper understanding of Cajatambo’s flora and its altitudinal distribution, contributing significantly to the knowledge of plant diversity in this understudied region of the Peruvian Andes and establishing a solid foundation for future ecological and conservation studies in the area.

## Figures and Tables

**Figure 1 plants-13-03328-f001:**
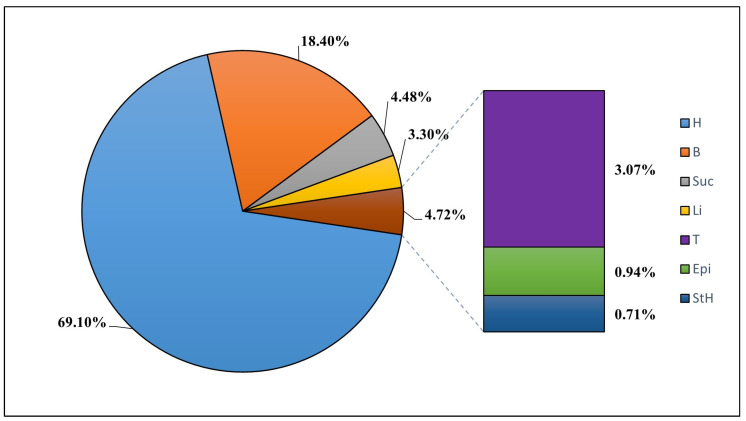
Species observed in the altitudinal gradient of Cajatambo. The following abbreviations are used for life forms (plant’s growth habit): H: herbaceous; B: blush; Suc: succulent; Li: liana; T: tree; Epi: epiphytes; StH: stoloniferous herbaceous.

**Figure 2 plants-13-03328-f002:**
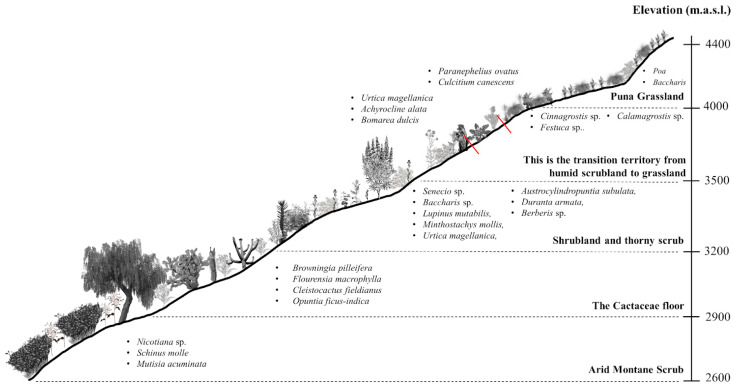
Vegetation profile along an altitudinal gradient in Cajatambo, Peru.

**Figure 3 plants-13-03328-f003:**
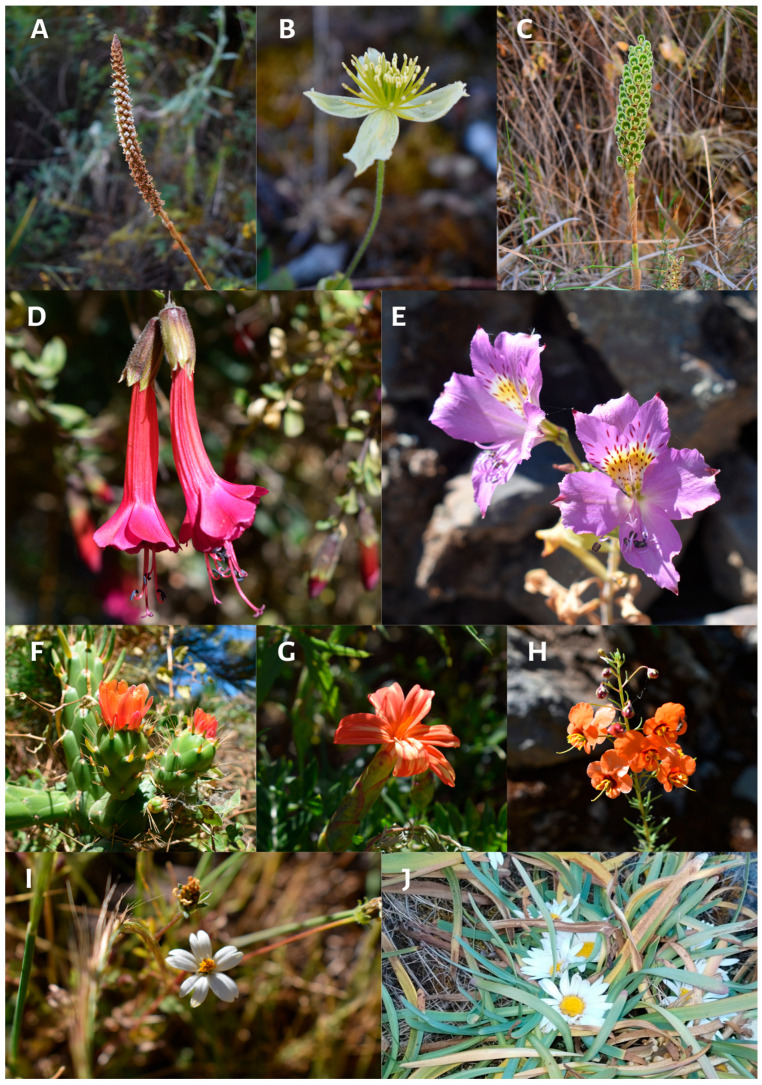
*Aa paleacea* (**A**), *Clematis peruviana* (**B**), *Altensteinia fimbriata* (**C**), *Cantua buxifolia* (**D**), *Alstroemeria lineatiflora* (**E**), *Austrocylindropuntia subulata* (**F**), *Mutisia acuminata* (**G**), *Alonsoa* sp. (**H**), *Cerastium* sp. (**I**), *Paranephelius ovatus* (**J**).

**Figure 4 plants-13-03328-f004:**
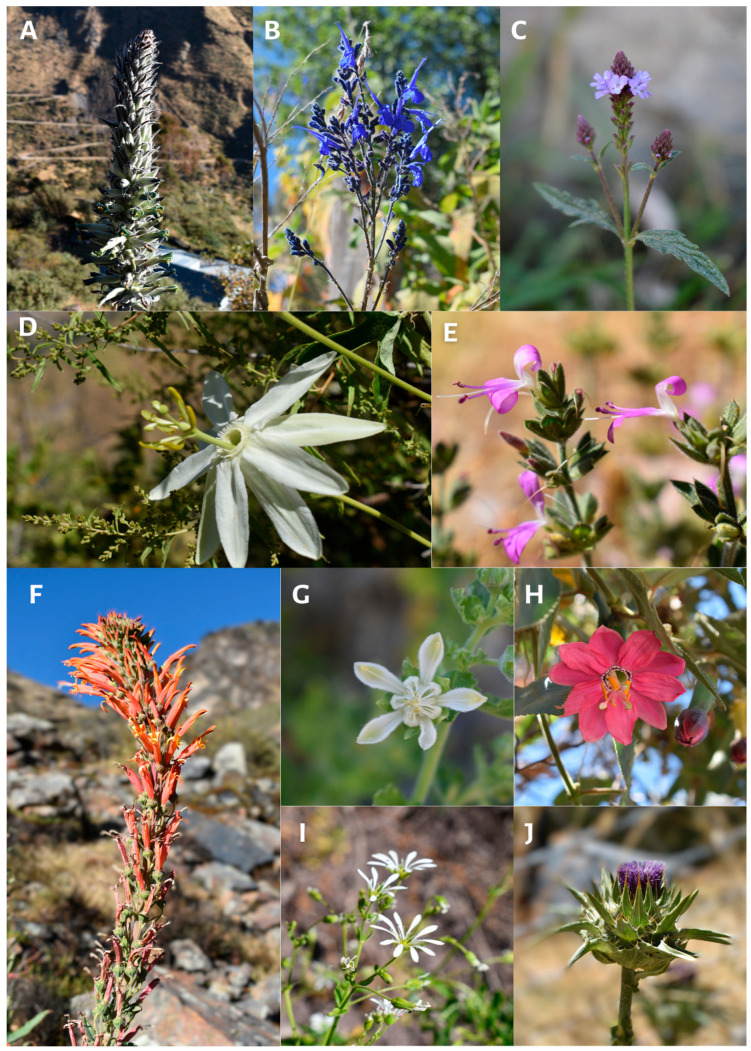
*Puya alpestris* (**A**), *Salvia sagittata* (**B**), *Verbena litoralis* (**C**), *Passiflora peduncularis* (**D**), *Dicliptera hookeriana* (**E**), *Siphocampylus tupaeformis* (**F**), *Presiliophytum incanum* (**G**), *Passiflora mixta* (**H**), *Arenaria* sp. (**I**), *Silybum marianum* (**J**).

**Figure 5 plants-13-03328-f005:**
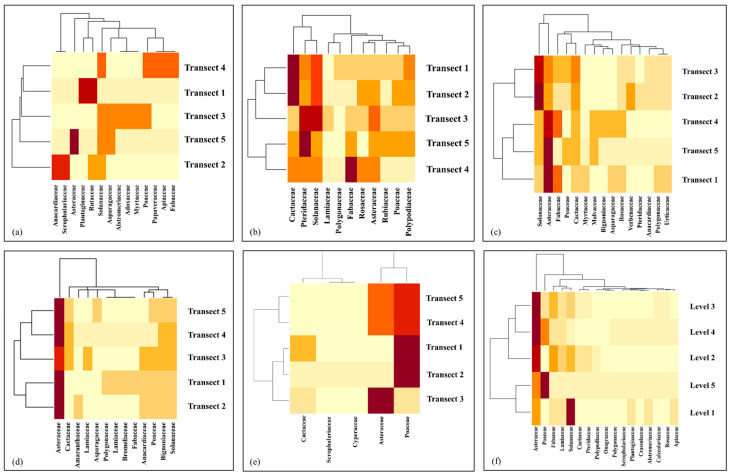
Heat map showing the results of a family composition analysis within arid montane scrub (**a**), Cactaceae floor (**b**), shrubland and thorny scrub (**c**), the transition territory from humid scrubland to grassland (**d**), puna grassland (**e**), and altitudinal gradient (**f**). A greater degree of darkness in the color tones indicates a higher number of species per family.

**Figure 6 plants-13-03328-f006:**
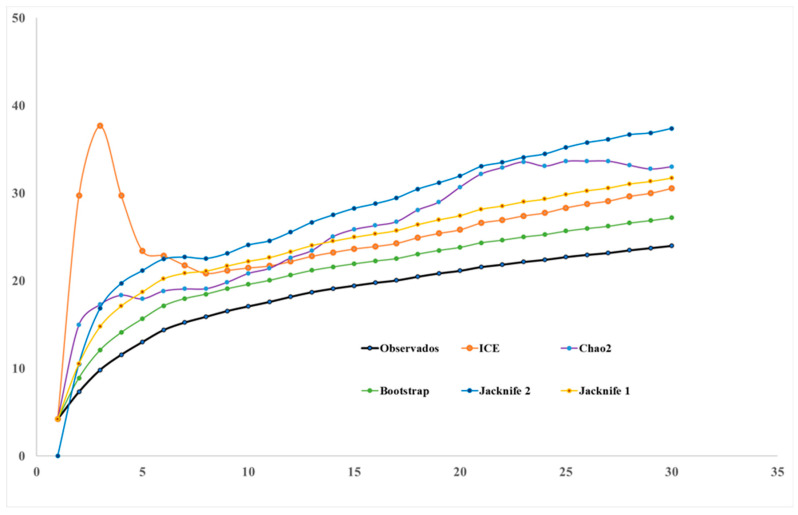
Species accumulation curves that define the efficiency of sampling in an altitudinal gradient in Cajatambo, Lima.

**Figure 7 plants-13-03328-f007:**
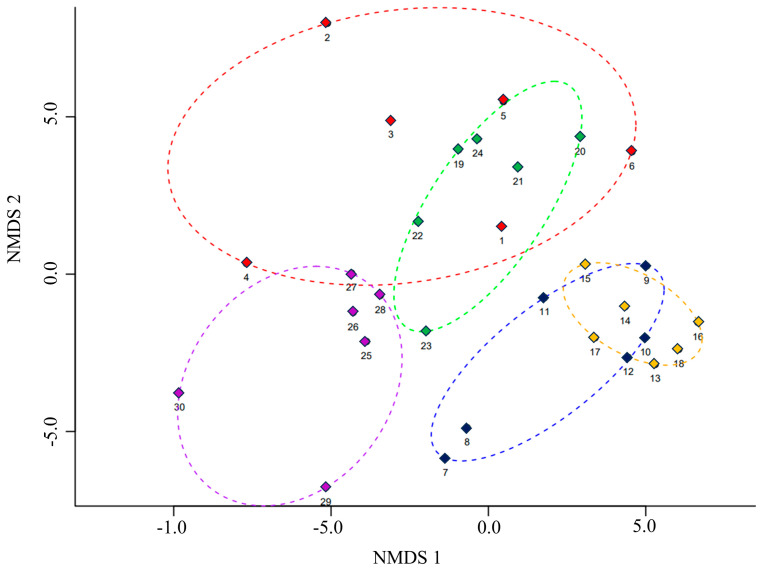
Multidimensional non-metric analysis (NMDS) for the abundance and richness of families in the five study areas (altitudinal gradient) in the Cajatambo district. The numbers represent the transect evaluated at each elevation level.

**Figure 8 plants-13-03328-f008:**
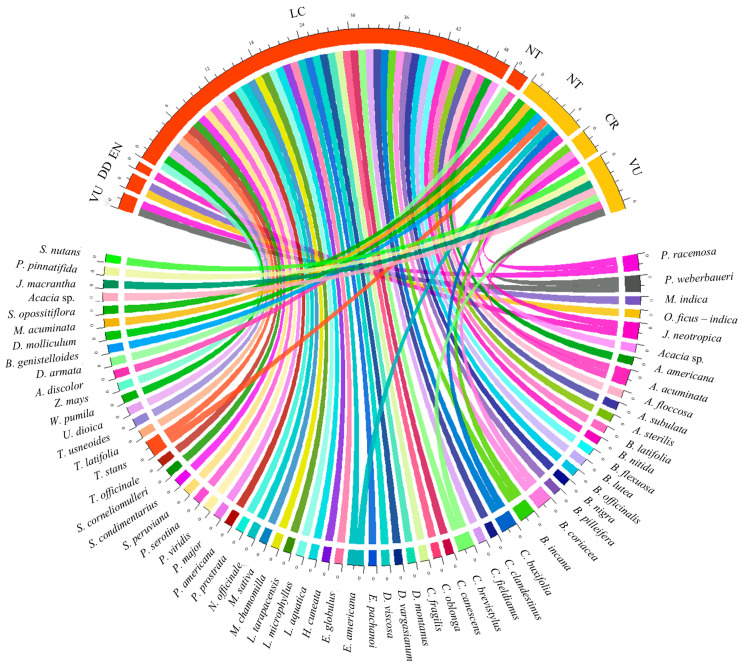
Threatened flora according to the UICN Red List and the Decreto Supremo N.° 043–2006–AG.

**Figure 9 plants-13-03328-f009:**
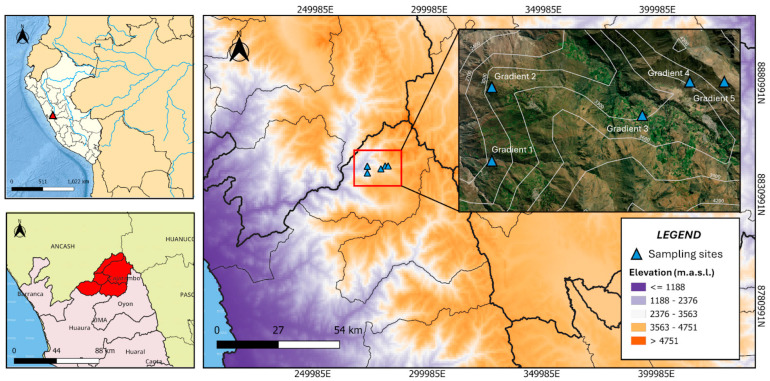
Location of the transects in the altitudinal gradient in the Cajatambo district.

**Table 1 plants-13-03328-t001:** Number of families, genera, and species of the main plant taxa in Cajatambo.

Taxa	Family	Genera	Species
Dicotyledon	56	191	337
Monocotyledon	9	35	60
Equisetophyta	1	1	1
Gnetophyta	1	1	1
Pteridophyta	8	14	25

**Table 2 plants-13-03328-t002:** List of species recorded in Cajatambo, Peru.

Family	Taxon	Plant’s Growth Habit	Conservation Status	Altitudinal Level
Acanthaceae	*Dicliptera hookeriana* Nees	H		2,3
Adoxaceae	*Sambucus peruviana* Kunth.	T	LC ^a^	1,2,3
Alstroemeriaceae	*Alstroemeria lineatiflora* Ruiz & Pav.	H		3
	*Bomarea dulcis* (Hook.) Beauverd	Li		2,3
	*Bomarea ovata (Cav.)* Mirb.	Li		1,2,3,4
	*Bomarea* sp. 1	Li		4,5
	*Bomarea* sp. 2	Li		4,5
	*Bomarea* sp. 3	Li		4,5
Amaranthaceae	*Alternanthera lanceolata* (Benth.) Schinz	H		1,2
	*Alternanthera macbridei* Standl.	H		3
	*Dysphania ambrosioides* L.	H		1,2,3
Anacardiaceae	*Mangifera indica* L.	T	DD ^a^	1,2,3
	*Schinus molle* L.	T		1,2,3,4
Apiaceae	*Bowlesia sodiroana* H.Wolff	H		2,3,4
	*Conium maculatum* L.	H		2,3,4
	*Daucus montanus* Humb. & Bonpl. ex Spreng.	H	LC ^a^	2,3
	*Eremocharis longiramea* (H. Wolff) I.M. Johnst	H		3,4
	*Eryngium humile* Cav.	H		4
	*Foeniculum vulgare* Mill.	H		1,2,3
	*Niphogeton stricta* (H. Wolff) Mathias & Constance	H		3,4
	*Oreomyrrhis andicola* (Kunth) Endl. ex Hook. f.	H		4
Asparagaceae	*Agave americana* L.	StH	LC ^a^	1,2,3
	*Furcraea occidentalis* Trel.	StH		1,2,3
Aspleniaceae	*Asplenium peruvianum* Desv.	H		2,3
	*Asplenium triphyllum* C. Presl	H		2,3
Asteraceae	*Achyrocline alata* (Kunth) DC.	H		3,4,5
	*Achyrophorus taraxacoides* Walp.	H		3
	*Ageratina azangaroensis* (Sch. Bip. ex Wedd.) R.M. King & H. Rob.	H		3,4
	*Ageratina glechonophylla* (Less.) R.M. King & H. Rob.	H		3,4
	*Ageratina pentlandiana* (DC.) R.M. King & H. Rob.	H		3,4
	*Ageratina pichinchensis* (Kunth.) R.M. King & H. Rob.	H		3,4,5
	*Ageratina* sp.	H		4,5
	*Ageratina sternbergiana* (DC.) R.M. King & H. Rob.	H		3,4
	*Xanthium fruticosum* (Medik.) L. fil.	B		2,3,4
	*Ambrosia cumanensis* Kunth	H		3,4
	*Aristeguietia discolor* (DC). R.M. King & H. Rob.	H	NT ^a^	3,4,5
	*Baccharis alpina* Kunth	B		3,4,5
	*Baccharis caespitosa* (Ruiz & Pav.) Pers.	B		4,5
	*Baccharis genistelloides* (Lam.) Pers.	B	NT ^b^	4,5
	*Baccharis latifolia* (Ruiz & Pav.) Pers.	B	LC ^a^	2,3,4
	*Baccharis nitida* (Ruiz & Pav.) Pers.	B	LC ^a^	3,4
	*Baccharis odorata* Kunth	B		4,5
	*Baccharis salicina* Torr. & A. Gray	B		3,4
	*Baccharis* sp. 1	B		3,4
	*Baccharis* sp. 2	B		3
	*Baccharis tola* subsb. Tola	B		4,5
	*Baccharis tricuneata* (L.f.) Pers.	B		3,4,5
	*Barnadesia horrida* Muschl.	B		4
	*Belloa* sp.	H		4,5
	*Bidens andicola* Kunth	H		2,3,4
	*Calendula officinalis* L.	H		2,3
	*Chaptalia cordata* Hieron.	H		3,4
	*Chaptalia nutans* (L.) Polák	H		4
	*Chersodoma antennaria* (Wedd.) Cabrera	H		4
	*Laennecia artemisiifolia* (Meyen & Walp.) G.L. Nesom	H		2,3
	*Conyza bonariensis* (L.) Cronquist	H		2,3
	*Conyza canadensis* (L.) Cronquist	H		2,3
	*Conyza sumatrensis* (Retz.) E. Walker	H		2,3
	*Coreopsis fasciculata* var. laevigata Sherff	H		2,3,4
	*Cosmos* sp.	H		3,4
	*Cronquistianthus determinatus* (B.L. Rob.) R.M. King & H. Rob.	B		2,3
	*Cronquistianthus lavandulifolius* (DC.) R.M. King & H.Rob.	B		2,3
	*Cronquistianthus macbridei* R.M. King & H. Rob.	B		2,3
	*Cronquistianthus* sp.	B		2,3
	*Flourensia macrophylla* S.F. Blake	B		2,3
	*Gnaphalium dombeyanum* DC.	H		4
	*Gynoxys nitida* Muschl.	B		4
	*Gynoxys* sp.	B		4
	*Heliopsis buphthalmoides* (Jacq.) Dunal	H		2,3,4
	*Heliopsis* sp.	H		2,3
	*Hieracium leptocephalum* Benth.	H		2,4
	*Hypochaeris chillensis* (Kunth) Britton	H		2,3
	*Jungia axillaris* (Lag. ex DC.) Spreng	B		2,3
	*Jungia paniculata* A. Gray	B		2,3
	*Lasiocephalus* sp.	H		3
	*Lomanthus arnaldii* (Cabrera) B. Nord. & Pelser	H		3,4
	*Andicolea ferruginea* (Ruiz & Pav.) Wedd.	B		4
	*Matricaria chamomilla* L.	H	LC ^a^	2,3
	*Mutisia acuminata* Ruiz & Pav.	B	NT ^b^	1,2,3,4
	*Onoseris odorata* Hook & Arn.	H		2,3
	*Ophryosporus chilca* (Kunth) Hieron.	B		2,3
	*Ophryosporus heptanthus* (Sch. Bip. ex Wedd.) R.M. King & H. Rob.	B		2,3
	*Ophryosporus peruvianus* R.M. King & H. Rob.	B		2,3
	*Ophryosporus piquerioides* (DC.) Benth. ex Baker	B		2,3
	*Pappobolus* sp.	H		2,3
	*Paranephelius ovatus* A. Gray ex Wedd.	H		4
	*Perezia multiflora* (Humb. & Bonpl.) Less.	H		4
	*Perezia pinnatifida* (Humb. & Bonpl.) Wedd.	H	Vu ^b^	4
	*Philoglossa peruviana* DC.	H		3,4
	*Senecio adenophyllus* Meyen & Walp	H		3,4
	*Senecio bangii* Rusby	H		3,4
	*Senecio pectioides* Rusby	H		3,4
	*Culcitium canescens* (Humb. & Bonpl.) Cuatrec.	H	LC ^a^	4
	*Senecio collinus* DC.	H		4
	*Senecio comosus* var. culcitioides (Sch.Bip.) Cabrera	H		3,4
	*Senecio condimentarius* Cabrera	H	LC ^a^	4
	*Senecio crassiflorus* DC.	H		4
	*Senecio evacoides* Sch.Bip. ex Wedd.	H		3
	*Senecio ferreyrae* Cabrera	H		3
	*Senecio genisianus* Cuatrec.	H		3,4
	*Senecio hastatifolius* Cabrera	H		3,4
	*Senecio hebetatus* Wedd.	H		3,4
	*Senecio minesinus* Cuatrec.	H		3
	*Senecio nutans* Sch. Bip.	H	Vu ^b^	4
	*Senecio sinapoides* Rusby	H		3,4
	*Senecio sulinicus* Cabrera	H		4
	*Lomanthus tovarii* (Cabrera) B.Nord. & Pelser	H		3
	*Silybum marianum* (L.) Gaertn.	H		2,3
	*Simsia dombeyana* DC.	H		3
	*Simsia* sp.	H		3
	*Sonchus asper* (L.) Hill	H		3,4
	*Sonchus oleraceus* (L.) L.	H		2,3,4
	*Tagetes elliptica* Sm.	H		2,3,4
	*Tagetes filifolia* Lag.	H		2,3
	*Tagetes minuta* L.	H		2,3
	*Tagetes multiflora* Kunth	H		3,4
	*Taraxacum officinale* F.H. Wigg	H	LC ^a^	1,2,3,4
	*Rockhausenia nubigena* Kunth	H		4
	*Werneria pumila* Kunth	H	LC ^a^	4
	*Werneria villosa* A. Gray	H		4,5
	*Xanthium spinosum* L.	H		3
	*Xenophyllum* sp.	H		4
Bassellaceae	*Ullucus* sp.	H		2,3
	*Ullucus tuberosus* Caldas	H		2,3
Berberidaceae	*Berberis flexuosa* Ruiz & Pav.	B	LC^a^	2,3
	*Berberis lutea* Ruiz & Pav.	B	LC^a^	2,3
	*Berberis* sp.	B		2,3
Betulaceae	*Alnus acuminata* Kunth	T	LC ^a^,Vu ^b^	2,3
Bignoniaceae	*Tecoma stans* (L.) Juss. ex Kunth	T	LC ^a^, NT ^b^	3,4
Boraginaceae	*Borago officinalis* L.	H	LC ^a^	4
Brassicaceae	*Brassica nigra* (L.) K. Koch	H	LC ^a^	2,3
	*Brassica rapa* L.	H		2,3
	*Capsella* sp.	H		3
	*Coronopus didymus* (L.) Sm.	H		3
	*Lepidium bipinnatifidum* Desv.	H		3,4
	*Lepidium chichicara* Desv.	H		3,4
	*Nasturtium officinale* W. T. Aiton	H	LC ^a^	2,3
	*Raphanus raphanistrum* subsp. sativus (L.) Domin	H		2,3
	*Weberbauera peruviana* (DC.) Al-Shehbaz	H		3,4
	*Weberbauera spathulifolia* (A. Gray) OE Schulz	H		3,4
Bromeliaceae	*Puya alpestris* (Poepp.) Gay			4
	*Tillandsia interrupta* Mez	Epi		3
	*Tillandsia latifolia* Meyen	Epi	LC ^a^	3,4
	*Tillandsia* sp.	Epi		2,3
	*Tillandsia usneoides* (L.) L.	Epi	LC ^a^	3,4
Cactaceae	*Austrocylindropuntia floccosa* (Salm-Dyck ex Winterfeld) F. Ritter	Suc	LC ^a^	4,5
	*Austrocylindropuntia subulata* (Muehlenpf.) Backeb.	Suc	LC ^a^	2,3,4
	*Browningia pilleifera* (F. Ritter) Hutchison	Suc	LC ^a^	2,3
	*Cleistocactus fieldianus* (Akers & Buining) Ostolaza	Suc	LC ^a^	2,3
	*Corryocactus brevistylus* (K.Schum. ex Vaupel) Britton & Rose	Suc	LC ^a^, Vu ^b^	3
	*Cumulopuntia galerasensis* F. Ritter	Suc		2,3
	*Echinopsis pachanoi* (Britton & Rose) Friedrich & G.D. Rowley	Suc	LC ^a^	2,3
	*Loxanthocereus* sp.	Suc		2,3
	*Opuntia ficus-indica* (L.) Mill.	Suc	DD ^a^	2,3
	*Oreocereus piscoensis* (Rauh & Backeb.) F. Ritter	Suc		2,3
Calceolariaceae	*Calceolaria annua* Edwin	H		3
	*Calceolaria aurea* Pennell	H		3
	*Calceolaria cajabambae* Kraenzl.	H		3,4
	*Calceolaria cuneiformis* subsp cuneiformis Ruiz & Pav.	H		3,4
	*Calceolaria glauca* Ruiz & Pav.	H		3,4
	*Calceolaria hispida* Benth.	H		2,3
	*Calceolaria parvifolia* Wedd.	H		3
	*Calceolaria* sp.	H		3,4
	*Porodittia triandra* (Cav.) G. Don	B		2,3
	*Calceolaria incarum* Kraenzl.	H		3,4
Campanulaceae	*Lobelia decurrens* Cav.	H		2,3
	*Siphocampylus* sp.	H		3,4
	*Siphocampylus tupiformis* Zahlbr.	H		3,4
Caryophyllaceae	*Arenaria* sp.	H		2,3
	*Cerastium* sp.	H		2,3
	*Drymania* sp.	H		3
	*Stellaria cuspidata* Willd. ex D.F.K. Schltdl.	H		2,3
	*Stellaria ovata* Willd. ex Schltdl.	H		2,3
Commelinaceae	*Commelina fasciculata* Ruiz & Pav.	H		2,3
	*Commelina* sp.	H		2,3
Convolvulaceae	*Cuscuta* sp.	H		2,3
Crassulaceae	*Echeveria andicola* pino	Suc		3,4
	*Echeveria chiclensis* (Ball) Berger	Suc		3,4
	*Echeveria excelsa* A.Berger CF	Suc		3,4
	*Echeveria wurdackii* Hutchison ex Kimnach cf	Suc		2,3,4
	*Sedum reniforme* (H. Jacobsen) Thiede & t‘ Hart	H		2,3
Cucurbitaceae	*Apodanthera* sp.	Li		3
	*Cucurbita maxima* Duchesne.	Li		2,3
	*Cyclanthera pedata* (L.) Schrad.	Li		2,3
Cyperaceae	*Eleocharis albibracteata* Nees & Meyen ex Kunth	H		4,5
Cystopteridaceae	*Cystopteris fragilis* (L.) Bernh.	H	LC ^a^	2,3
Dryopteridaceae	*Polystichum cochleatum* (Klotzsch) Hieron.	H		2,3
	*Polystichum montevidense* (Spreng.) Rosenst.	H		2,3
	*Polystichum orbiculatum* (Desv.) J. Rémy & Fée	H		3
Ephedraceae	*Ephedra americana* Humb. & Bonpl. ex Willd.	H	LC ^a^, NT ^b^	2,3,4
Equisetaceae	*Equisetum bogotense* Kunth	H		3,4,5
Ericaceae	*Pernettya prostrata* (Cav.) DC.	B	LC ^a^	3
Euphorbiaceae	*Croton ruizianus* Müll.Arg.	H		3,4
	*Jatropha macrantha* Müll. Arg.	B	Vu ^b^	3,4
	*Ricinus communis* L.	B		3
Fabaceae	*Acacia* sp.	T	LC ^a^	2,3,4
	*Astragalus garbancillo* Cav.	H		2,3
	*Astragalus* sp.	H		2,3
	*Caesalpinia spinosa* (Molina) Kuntze	T	Vu ^b^	2,3,4
	*Dalea cylindrica* Hook.	B		2,3
	*Desmodium mexicanum* Sweet	H		2,3
	*Desmodium molliculum (Kunth) DC.*	H	NT ^b^	2,3
	*Desmodium vargasianum* B.G. Schub.	H	LC ^a^	2,3
	*Indigofera* sp.	H		2,3,4
	*Lupinus brachypremnon* C.P. Sm.	B		3,4
	*Lupinus carazensis* Ulbr.	B		3,4
	*Lupinus condensiflorus* C.P. Sm.	B		3,4
	*Lupinus exochus* C.P. Sm.	B		3,4
	*Lupinus microphyllus* Desr.	B	LC ^a^	3,4
	*Lupinus mutabilis* Sweet	B		3,4
	*Lupinus* sp.	B		3,4
	*Lupinus tarapacensis* C.P.Sm.	B	LC ^a^	3,4
	*Medicago polymorpha* L.	H		3
	*Medicago sativa* L.	H	LC ^a^	2,3,4
	*Melilotus indicus* (L.) All.	H		2,3
	*Otholobium pubescens* (Poir.) J.W. Grimes	B		2,3
	*Psorallea* sp.	B		2,3
	*Senna birostris* (Dombey ex Vogel) H.S. Irwin & Barneby	B		2,3
	*Senna versicolor* (Vogel) H.S. Irwin & Barneby	B		2,3
	*Spartium junceum* L.	B		1,2,3,4
	*Vicia faba* L.	H		2,3
	*Vigna* sp.	H		2,3
Gentianaceae Geraniaceae	*Gentianella chrysotaenia* (Gilg) Zarucchi	H		4
	*Gentianella* sp. 1	H		4
	*Gentianella* sp. 2	H		4
	*Erodium cicutarium* (L.) L‘Hér.	H		2,3,4
	*Geranium ayacuchense* R. Knuth	H		3,4
	*Geranium chilloense* Willd. ex Kunth	H		3,4
	*Geranium herrerae* R. Knuth	H		3,4
	*Geranium* sp.	H		3
Hydrophyllaceae	*Phacelia secunda* J.F. Gmel.	H		3
Iridaceae	*Sisyrinchium junceum* E. Mey. ex C. Presl	H		4
Juglandaceae	*Juglans neotropica* Diels	T	EN ^a^, NT ^b^	2,3
Juncaceae	*Luzula racemosa* Desv.	H		4,5
Lamiaceae	*Clinopodium pulchellum* (Kunth) Govaerts	B		3,4
	*Clinopodium sericeum* (C.Presl ex Benth.) Govaerts	B		3,4
	*Clinopodium speciosum* (Hook.) Govaerts	B		3,4
	*Hyptis carpinifolia* Benth.	H		2,3,4
	*Leonotis nepetifolia* (L.) R. Br.	H		2,3
	*Lepechinia meyenii* (Walp.) Epling	H		3,4
	*Marrubium vulgare* L.	H		2,3,4
	*Melissa officinalis* L.	H		1,2,3
	*Mentha x piperita* var. citrata (Ehrh.) Briq.	H		2,3
	*Minthostachys mollis* (Benth.) Griseb.	B		2,3,4
	*Minthostachys* sp. 1	B		2,3,4
	*Minthostachys* sp. 2	B		4
	*Minthostachys* sp. 3	B		2,3,4
	*Origanum vulgare* L.	H		2,3
	*Salvia cuspidata* Ruiz & Pav.	H		2,3
	*Salvia opossitiflora* Ruiz & Pav.	H	NT ^b^	2,3
	*Salvia rhombifolia* Ruiz & Pav.	H		2
	*Salvia sagittata* Ruiz & Pav.	H		2,3
	*Salvia sarmentosa* Epling	H		3,4
	*Stachys pusilla* (Wedd.) Briq.	H		4
Lauraceae	*Persea americana* Mill.	T	LC ^a^	2,3
Loasaceae	*Cajophora cirsiifolia* C. Presl.	H		3,4
	*Cajophora* sp.	H		4
	*Nasa chenopodiifolia* (Desr.) Weigend	H		4
	*Nasa cymbopetala* (Urb. & Gilg) Weigend	H		4
	*Nasa magnifica* (Urb. & Gilg) Weigend	H		4
	*Presliophytum incanum* (Graham) Weigend	H		4
Loranthaceae	*Tristerix pubescens* Kuijt	B		3,4
Malvaceae	*Acaulimalva* sp.	H		2,3
	*Fuertesimalva peruviana* (L.) Fryxell	H		2,3
	*Malva* sp.	H		3
Montiaceae	*Claytonia humboldtiana* Kunth	H		3,4
Myrtaceae	*Eucalyptus globulus* Labill.	T	LC ^a^	1,2,3,4
Onagraceae	*Ludwigia* sp.	H		4
	*Oenothera grandis* Smyth	H		2,3
	*Oenothera laciniata* Hill	H		2,3
	*Oenothera multicaulis* Ruiz & Pav.	H		2,3,4
	*Oenothera rosea* L‘Hér. ex Aiton	H		2,3,4
Orchidaceae	*Aa paleacea* (Kunth) Rchb. f.	H		3,4
	*Altensteinia fimbriata* Kunth	H		3,4
Orobanchaceae	*Bartsia* sp.	H		3
Oxalidaceae	*Oxalis corniculata* L.	H		3,4
	*Oxalis megalorrhiza* Jacq	H		3
	*Oxalis* sp.	H		3,4
	*Oxalis tuberosa* Molina	H		2,3
Papaveraceae	*Argemone mexicana* L.	H		1,2,3
	*Argemone subfusiformis* Ownbey	H		2,3
Passifloraceae	*Passiflora mixta* L. f.	Li		3,4
	*Passiflora peduncularis* Cav.	Li		3,4
	*Passiflora* sp.	Li		3,4
	*Passiflora trifoliata* Cav.	Li		2,3,4
	*Passiflora tripartita* (Juss.) Poir.	Li		4
Piperaceae	*Peperomia galioides* Kunth	Suc		2,3,4
	*Peperomia glabella* (Sw.) A. Dietr.	Suc		4
	*Peperomia inaequalifolia* Ruiz & Pav.	Suc		3,4
	*Peperomia microphylla* Kunth	Suc		4
	*Peperomia trullaefolia* Trel	Suc		3.4
Plantaginaceae	*Plantago australis* Lam.	H		2,3,4
	*Plantago lamprophylla* Pilg.	H		4
	*Plantago lanceolata* L.	H		3,4
	*Plantago major* L.	H	LC ^a^	1,2,3
	*Veronica persica* Poir.	H		3,4
Poaceae	*Agrostis breviculmis* Hitchc.	H		3,4,5
	*Agrostis sodiroana* Hack.	H		4,5
	*Agrostis tolucensis* Kunth	H		4,5
	*Avena barbata* Pott ex Link	H		4,5
	*Avena sterilis* L.	H	LC ^a^	3,4
	*Bothriochloa saccharoides* (Sw.) Rydb.	H		3,4
	*Bromus catharticus* Vahl	H		4,5
	*Calamagrostis* sp. 1	H		3,4
	*Calamagrostis* sp. 2	H		4
	*Calamagrostis* sp. 3	H		4
	*Cenchrus clandestinus* (Hochst. ex Chiov.) Morrone	H	LC ^a^	4,5
	*Cinnagrostis brevifolia* (J. Presl) P.M. Peterson, Soreng, Romasch. & Barberá	H		4,5
	*Cinnagrostis macrophylla* (Pilg.) P.M. Peterson, Soreng, Romasch. & Barberá	H		3
	*Cinnagrostis recta* (Kunth) P.M. Peterson, Soreng, Romasch. & Barberá	H		4,5
	*Cinnagrostis rigescens* (J. Presl) P.M. Peterson, Soreng, Romasch. & Barberá	H		4,5
	*Cinnagrostis vicunarum* (Wedd.) P.M. Peterson, Soreng, Romasch. & Barberá	H		4,5
	*Cortaderia jubata* (Lemoine ex Carrière) Stapf	H		4,5
	*Dactylis glomerata* L.	H		4,5
	*Elymus angustatus* Steud.	H		3
	*Festuca dolichophylla* J. Presl	H		4,5
	*Festuca humilior* Nees & Meyen	H		4,5
	*Festuca weberbaueri* Pilg.	H		4,5
	*Hordeum vulgare* L.	H		2,3
	*Jarava ichu* Ruiz & Pav.	H		3,4
	*Lolium multiflorum* Lam.	H		2,3
	*Muhlenbergia peruviana* (P. Beauv.) Steud.	H		4,5
	*Nassella brachyphylla* (Hitchc.) Barkworth	H		3
	*Nassella mucronata* (Kunth) R.W. Pohl	H		4,5
	*Nassella soukupii* (Tovar) Barkworth	H		4,5
	*Paspalum candidum* (Humb. & Bonpl. ex Flüggé) Kunth	H		4,5
	*Poa brevis* Hitchc.	H		4,5
	*Poa* cf. *asperiflora* Hack	H		4,5
	*Poa fibrifera* Pilg.	H		4,5
	*Poa horridula* Pilg.	H		4,5
	*Polypogon interruptus* Kunth	H		4,5
	*Polypogon viridis* (Forssk.) Hyl.	H	LC ^a^	4,5
	*Schizachyrium cirratum* (Hack.) Wooton & Standl.	H		3,5
	*Stipa ichu* (Ruiz & Pav.) Kunth	H		3,4,5
	*Triticum sativum* Lam.	H		2,3
	*Zea mays* L.	H	LC ^a^	2,3
Polemoniaceae	*Cantua buxifolia* Juss. ex Lam.	B	LC ^a^, NT ^b^	2,3
Polygalaceae	*Monnina salicifolia* Ruiz & Pav.	H		3,4
	*Monnina* sp.	H		2,3
	*Pteromonnina macrostachya* (Ruiz & Pav.) B. Eriksen	H		2,3
Polygonaceae	*Muehlenbeckia volcanica* (Benth.) Endl.	H		3,4
	*Rumex acetosella* L.	H		2,3,4
	*Rumex crispus* L.	H		2,3,4
	*Rumex obtusifolius* L.	H		2,3,4
Polypodiaceae	*Campyloneurum* sp.	H		2,3
	*Pleopeltis* cf. *thyssanolepis*	H		2,3,4
	*Pleopeltis pycnocarpa* (C. Chr.) A.R. Sm.	H		2,3,4
	*Polypodium* sp.	H		2,3
	*Thelypteris caucaensis*	H		2,3
Portulacaceae	*Calandrinia ciliata* (Ruiz & Pav.) DC.	H		2,3
Pteridaceae	*Adiantum andicola* Liebm.	H		2,3
	*Adiantum poiretii* Wikstr.	H		2,3
	*Adiantum raddianum* C. Presl	H		2,3
	*Adiantum subvolubile* Mett.	H		3
	*Cheilanthes bonariensis* (Willd.) Proctor	H		2,3
	*Cheilanthes microphylla* (Sw.) Sw.	H		2,3
	*Cheilanthes pilosa* Goldm.	H		2,3,4
	*Cheilanthes pruinata* Kaulf.	H		2,3,4
	*Notholaena nivea* (Poir.) Desv	H		2,3,4
	*Pellaea ternifolia* (Cav.) Link	H		2,3
	*Pityrogramma ferruginea* (Kunze) Maxon	H		2,3
Ranunculaceae	*Clematis peruviana* DC.	Li		3,4
	*Clematis seemannii* Kuntze	B		2,3
	*Ranunculus praemorsus* Kunth ex DC.	H		3,4
Rosaceae	*Alchemilla pinnata* Ruiz & Pav.	H		3,4
	*Cydonia oblonga* Mill.	B	LC ^a^	2
	*Hesperomeles cuneata* Lindl.	B	LC ^a^	2
	*Malus domestica* Borkh.	B		2,3
	*Margyricarpus pinnatus* (Lam.) Kuntze	B		3,4
	*Polylepis racemosa* Ruiz & Pav.	T	Vu ^a^, CR ^b^	3
	*Polylepis weberbaueri* Pilg.	T	Vu ^a,b^	4
	*Prunus serotina* Ehrh.	B	LC ^a^	2,3
	*Rubus* sp.	B		3,4
	*Tetraglochin cristatum* (Britton) Rothm.	B		3
Rubiaceae	*Arcytophyllum* sp.	H		2,3
	*Arcytophyllum thymifolium* (Ruiz & Pav.) Standl.	B		2,3
	*Galium aparine* L.	H		2,3
	*Galium corymbosum* Ruiz & Pav.	H		2,3
Rutaceae	*Ruta chalepensis* L.	H		1,2,3
	*Ruta graveolens* L.	H		1,2,3
Sapindaceae	*Dodonaea viscosa* (L.) Jacq.	H	LC ^a^	2,3
Saxifragaceae	*Saxifraga* sp.	H		3
Scrophulariaceae	*Alonsoa acutifolia* Ruiz & Pav.	H		3,4
	*Alonsoa linearis* (Jacq.) Ruiz & Pav.	H		3,4
	*Alonsoa meridionalis* (L.f.) Kuntze	H		3,4
	*Alonsoa* sp.	H		3,4
	*Buddleja coriacea* Remy	B	LC ^a^, CR ^b^	4
	*Buddleja incana* Ruiz & Pav.	T	LC ^a^, CR ^b^	4
	*Limosella aquatica* L.	H	LC ^a^	2,3
Selaginellaceae	*Selaginella* sp.	H		2,3
Solanaceae	*Cestrum auriculatum* L‘Hér.	B		3,4
	*Datura stramonium* L.	H		3
	*Dunalia obovata* (Ruiz & Pav.) Dammer	B		2,3
	*Dunalia spinosa* (Meyen) Dammer.	B		2,3
	*Jaltomata bicolor* (Ruiz & Pav.) Mione.	B		2,3
	*Jaltomata* sp.	B		2,3
	*Lycopersicum* sp.	H		1,2,3
	*Nicandra physalodes* (L.) Gaertn.	H		2,3
	*Nicotiana glutinosa* L.	H		1,2,3
	*Nicotiana paniculata* L.	H		1,2,3
	*Nicotiana rustica* L.	H		1,2,3
	*Physalis peruviana* L.	H		1,2,3
	*Salpichroa* sp. 1	B		2,3
	*Salpichroa* sp. 2	B		2,3
	*Salpichroa weberbaueri* Dammer	B		2,3
	*Solanum candolleanum* Berthault	H		2,3
	*Solanum corneliomulleri* J.F. Macbr.	H	LC ^a^	3,4
	*Solanum pentlandii* Dunal	H		2
	*Solanum* sp. 1	H		4
	*Solanum* sp. 2	H		4
	*Solanum* sp. 3	H		4
	*Solanum* sp. 4	H		4
	*Solanum tuberosum* L.	H		3,4
Thelypteridaceae	*Thelypteris aff. patens* (Sw.) Small	H		2,3
Tropaeolaceae	*Tropaeolum tuberosum* Ruiz & Pav.	H		2,3
Urticaceae	*Urtica dioica* L.	H	LC ^a^	2,3
	*Urtica magellanica* Juss. ex Poir.	H		2,3,4
	*Urtica urens* L.	H		2,3
Valerianaceae	*Valeriana agrimonifolia* Killip	H		3,4
Verbenaceae	*Aloysia triphylla* Royle	H		2,3
	*Duranta armata* Moldenke	B	NT ^a^	2,3
	*Lantana* sp.	B		2,3
	*Verbena litoralis* Kunth	H		2,3
Woodsiaceae	*Woodsia montevidensis* (Spreng.) Hieron.	H		3,4

Reported species for plant´s growth habit: H: herbaceous; B: blush; Suc: succulent; Li: liana; T: tree; Epi: epiphytes; StH: stoloniferous herbaceous; conservation status: DD: data deficient, LC: least concern, NT: near threatened, VU: vulnerable, EN: endangered, CR: critically endangered; ^a^ Red List IUCN, ^b^ Peruvian LSA N° 043–2006–AG; altitudinal level: levels 1 to 5.

## Data Availability

Data used in this study can be requested from the corresponding author via email: yquinterosg@unmsm.edu.pe.

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
