# Peer review of "Floristic Diversity and Distribution Pattern along an Altitudinal Gradient in the Central Andes: A Case Study of Cajatambo, Peru"

_plants, 2024, doi:10.3390/plants13233328_

Round 1
Reviewer 1 Report
Comments and Suggestions for Authors
The manuscript plants-3114629 (“Diversity and Distribution of Vegetation along an Altitudinal Gradient in the Central Andes: A Case Study of Cajatambo, Perú”) investigate flora species along an altitudinal gradient in the Cajatambo. The work is very interesting but I have doubts before publication.
I recommend to the authors change some words in the keyword. It’s better that no coincidence with the title words (p.e. vegetation)
The manuscript is correct and is good. The references are very good and the history very well explained in the introduction and discussion.
It is really difficult to understand the results if you have not previously read the methodology at the end of the document. I think it is more convenient to include the methodology section before the results to better understand the analysis. But I understand that these are journals rules.
With regard to the method, I think it is deficient. The authors talk about the number of plots and transects but at no point do they indicate what they do in each transect. I understand that they collect material for later identification but I don't know if they take vegetation cover, measurements of species, etc. The authors should write better the study method. Please clarify.
On the other hand, in data analysis I do not see the linear regression models that the authors indicated they did.
Some comments:
Line 185: In the table 2, I would like know why the authors did not use Raunkiaer clasification in the "Plant’s Growth habit"
Lines 223-229: The author could indicate the estimator valour for the ANOVA analysis (p.e. ANOVA, F=… p<0.0002) and regression models.
Lines 247-252: In this sections, It is be interesting show the percentage of how many species are include in each categories in the text (VU, DD, EN, LC, NT, CR).
In the discussion sections, in the first paragraph it is difficult to understand what the authors want to indicate (lines 260-264). Please clarify.
Line 288: The genera Agrostis, Calamagrostis and Poa in italic.
Line 369: Change “Figure 4” by “Figure 9”.
Reviewer 2 Report
Comments and Suggestions for Authors
The paper provides new information about vascular plant diversity and distribution pattern alon an altitutidnal gradient in the Central Andes, Peru. Results are based on the systematic fiedl surveys and detailed and quality analysis. Floristic charecteristics are clearly presented and compared with relevant data in literature sources. Topic of paper is very relevant to broader knowledge abotu plant and ecosystem diversity in the Central Andes, Peru. IT also contributes to planning the future conservation measures in the area.
Following are comments and suggestions for authors:
1) Row 2: consider to change title in: Floristic diversity and distrubution paatern along..., because "vegetation" in title suggest to reader that paper provides information about plant communities, but the focus of the survey is vascular plant diversity and distribution patterns of investigated flora species. 2) Row 49: "mainly angiosperms, gymnosperms and ferns", combined in "vascular plants" because any botanical expert and scientist knows definition of vascular plants. 3) 55-56, "the prevalence of endemic species" - it would be useful to readers to indicate number, how many endemic species there are in Peru, if needed add additional reference in Literature.; 4) Row 90: "Equisetophyta", in Table 1 there is "Sphenophyta", please correct the valid taxon name, 5) In Table 2, List of species, in column Family, after "Gentianella sp. 2" add name of the family Geraniacea, alligned with species name Erodium cicutarium; 6) Row 315-316 "model indicate that richness of Asteraceae species in the study area is regulated by the richness of Asteraceae species", please consider to rephrase this sentence, it sounds rather ambigous.
The English language is of high quality.
